# Antiplatelet Effects of PCSK9 Inhibitors in Primary Hypercholesterolemia

**DOI:** 10.3390/life11060466

**Published:** 2021-05-23

**Authors:** Piotr Pęczek, Mateusz Leśniewski, Tomasz Mazurek, Lukasz Szarpak, Krzysztof J. Filipiak, Aleksandra Gąsecka

**Affiliations:** 11st Chair and Department of Cardiology, Medical University of Warsaw, 00-927 Warsaw, Poland; piotrpeczek44@gmail.com (P.P.); mlesniewski76@gmail.com (M.L.); tmazurek@kardia.edu.pl (T.M.); krzysztof.filipiak@wum.edu.pl (K.J.F.); 2Department of Research Outcomes, Maria Sklodowska-Curie Medical Academy in Warsaw, 03-411 Warsaw, Poland; lukasz.szarpak@gmail.com; 3Maria Sklodowska-Curie Bialystok Oncology Center, Department of Research Outcomes, 15-027 Bialystok, Poland; 4Department of Cardiology, University Medical Center Utrecht, 3584 CX Utrecht, The Netherlands

**Keywords:** PCSK9 inhibitors, platelets, primary hypercholesterolemia, LDL-cholesterol, atherosclerosis, treatment

## Abstract

Proprotein convertase subtilisin-kexin type 9 (PCSK9) inhibitors are a novel group of hypolipidemic drugs that are recommended particularly for high-risk hypercholesterolemia patients, including those with primary hypercholesterolemia (PH), where lifelong exposure to high low-density lipoprotein (LDL) cholesterol levels results in an elevated risk of atherosclerosis at an early age. The onset and progression of atherosclerosis is significantly influenced by activated platelets. Oxidized LDL influences platelet activation by interacting with their surface receptors and remodeling the composition of their cell membrane. This results in platelet aggregation, endothelial cell activation, promotion of inflammation and oxidative stress, and acceleration of lipid accumulation in atherosclerotic plaques. PCSK9 inhibitors reduce platelet activation by both significantly lowering LDL levels and reducing the LDL receptor-mediated activation of platelets by PCSK9. They also work synergistically with other hypolipidemic and antithrombotic drugs, including statins, ezetimibe, acetylsalicylic acid, clopidogrel, and ticagrelor, which enhances their antiplatelet and LDL-lowering effects. In this review, we summarize the currently available evidence on platelet hyperreactivity in PH, the effects of PCSK9 inhibitors on platelets, and their synergism with other drugs used in PH therapy.

## 1. Introduction

Primary hypercholesterolemia (PH) is a metabolic disorder characterized by elevated serum levels of low-density lipoprotein cholesterol (LDL-C). This lipid disorder is genetically heterogenous and involves both monogenic autosomal dominant familial hypercholesterolemia (FH), with a prevalence estimated at 1:250 [1], and the more frequent polygenic non-familial hypercholesterolemia [2]. Numerous epidemiological studies have proved a correlation between serum LDL-C levels and cardiovascular disease (CVD) [3,4,5,6,7]. Due to lifelong exposure to high cholesterol levels, individuals with PH have a greater risk of developing CVD, even at a relatively young age [8,9].

Three main genes have been identified as causative factors of FH in an autosomal dominant manner: the LDL receptor (*LDLR*), apolipoprotein B (*ApoB*), and the proprotein convertase subtilisin-kexin type 9 (*PCSK9*) [10]. There are also other rare forms of FH. These include mutations in the apolipoprotein E (*ApoE*) gene and LDL-C adaptor protein 1 (*LDLRAP1*) [11,12]. Polygenic non-familial hypercholesterolemia includes single nucleotide polymorphisms (SNPs) in several genes, involving common genes (*LDLR, PCSK9*) as well as less frequent genes, such as the ATP-binding cassette sub-family G member 8 gene (*ABCG8*) or cadherin EGF LAG seven-pass G-type receptor 2 gene (*CELSR2*) [13]. Both *ABCG8* and *CELSR2* encode proteins associated with transmembrane transport and receptor–ligand cellular interactions [14,15].

PH is associated with increased platelet reactivity. Activated platelets play a key role in atherosclerotic processes and the interaction between platelets and oxidized LDL (oxLDL) affects the formation of atherosclerotic plaques in several ways, which are discussed in detail later [16,17].

Until the early 2000s, long-term lipoprotein apheresis was the only treatment to improve outcomes in patients with severe FH [18]. In 2003, PCSK9 was discovered in some families, presenting with the clinical phenotype of FH yet without pathogenic DNA variants in either the *LDLR* or *ApoB* genes [19]. At that time, only these two genes were known to cause FH, and so a new responsible gene was suspected. Further genetic research identified a region on chromosome 1 that was linked to the presence of this phenotype [20,21]. Eventually, in 2003, scientists found that mutations in the PCSK9 gene were able to cause FH in those patients [22]. Since then, PCSK9 has become a baseline for several therapeutic agents, which significantly reduce the risk of cardiovascular events [23]. Recent findings show that PCSK9 inhibitors may lower LDL-C level, as well as decrease platelet activity [24]. Other pleiotropic effects of PCSK9 inhibitors, such as anti-atherosclerotic effects, stabilization of atherosclerotic plaques, antineoplastic effects, and the ability to influence the course of bacterial infections, have recently been comprehensively reviewed [25].

In this review, we focus on the currently available evidence on platelet hyperreactivity in PH and the effects of PCSK9 inhibitors on platelets, including: (i) the pathophysiology of atherosclerosis in PH, (ii) the mechanisms underlying platelet hyperreactivity in PH, and (iii) the anti-atherogenic and antiplatelet effects of PCSK9 inhibitors.

## 2. Pathophysiology

### 2.1. Development of Atherosclerotic Plaques

Atherosclerosis is a complex process consisting of several steps. First, LDL particles cross the arterial endothelium and accumulate in the intima or subendothelial layer [26]. This step is determined by the integrity of the endothelium. Regions with turbulent blood flow, such as arterial bifurcations, are more vulnerable to this process [27]. Numerous genetic factors, oxidative and mechanical stress, elevated serum homocysteine levels, and infections also contribute to this process [17]. Once in the intima, LDL is oxidized, triggering the expression of various adhesion and chemoattractant particles, such as intercellular adhesion molecule-1 (ICAM-1), vascular cell adhesion molecule-1 (VCAM-1), platelet–endothelial cell adhesion molecule (PECAM-1), selectins, and integrins (CD11/CD18), driving the recruitment of macrophages to the site [28,29,30]. Within the arterial wall, macrophages begin to internalize ox-LDL via scavenger receptors, eventually transforming into foam cells [31]. This intensifies the ongoing inflammation [32], inducing the production and release of even more cytokines, which further promote the attraction of macrophages [33].

As the cycle repeats and additional lipids accumulate in the intima, a fibrous cap forms, composed mostly of a lipid-rich core and smooth muscle cell cap, which separates the atherosclerotic plaque from the blood flow [33]. The development of an atherosclerotic plaque is shown schematically in Figure 1.

### 2.2. Mechanisms of Platelet Activation in Hypercholesterolemia

Platelet activation in hypercholesterolemic states occurs through several mechanisms, including: (i) scavenger receptor cluster of differentiation (CD)36, (ii) scavenger receptor lectin-like ox-LDL receptor-1 (LOX-1), and (iii) LDL-C triggered platelet membrane composition changes [34]. The mechanisms of platelet activation by LDL are shown in Figure 2.

CD36 is a multi-functional class B scavenger receptor [35]. It is a transmembrane glycoprotein that is constitutively expressed in various cell types, including macrophages, platelets, and microvascular endothelial cells [35,36]. It is a ligand for a number of particles, such as thrombospondin-1, ox-LDL, fatty acids, microbial diacyloglycerides, and many others [35,37,38]. Previous studies have shown that the interaction of CD36 with ox-LDL triggers signaling pathways that activate platelets, inducing the expression of P-selectin and the activation of integrin α_IIb_β_3_ (the receptor for fibrinogen), therefore facilitating the formation of platelet–leukocyte complexes via P-selectin and the cross-linking of adjacent platelets via fibrinogen [34,39]. The ox-LDL–CD36 interaction was shown to trigger platelet hyperreactivity via Src family kinases, Vav-guanine nucleotide exchange factors, cyclic guanosine monophosphate (cGMP), and nicotinamide adenine dinucleotide phosphate (NADPH) oxidase, producing reactive forms of oxygen and leading to a vicious circle of LDL oxidation and platelet activation [40,41]. The binding of ox-LDL to CD36 also induces the release of various chemokines, such as monocyte chemotactic protein-1 and the interleukin 1β precursor, leading to the progression of atherosclerosis [42,43]. CD36 activation is also an important factor contributing to ox-LDL platelet internalization and foam cell formation [34,43].

Another platelet receptor that is important in the development of atherosclerosis is LOX-1. It is a class E scavenger receptor involved in the regulation of ox-LDL uptake by endothelial cells and platelets [44,45]. Contrary to the native expression of CD36, LOX-1 expression is atheroma-related [34,46]. Binding between ox-LDL and LOX-1 leads to the activation of integrins α_IIb_β_3_ and α_2_β_1_, which results in platelet shape change and aggregation, contributing to thrombus formation [44,45]. Ox-LDLs are also linked with high plasminogen activator inhibitor-1 levels and suppression of the fibrinolytic activity of endothelial cells [17].

Hypercholesterolemia can also contribute to platelet hyperreactivity via direct ox-LDL–platelet membrane interaction. As previous studies have shown, intrinsic platelet reactivity varies between individuals and increases with age [47]. LDL remodels the phospholipid composition of the platelet membrane by transferring phospholipids from lipoproteins, hence changing the structure of membrane phospholipids [48]. Therefore, high LDL-C levels activate platelets not only via intracellular signaling pathways, but also through direct lipid exchange [43].

### 2.3. Platelet Activation, Atherotogenesis, and Atherothrombosis

Platelet activation in hypercholesterolemia can promote thrombus formation on an injured artery, leading to arterial thrombosis [49], which is by far the most serious complication of atherosclerosis and can result in death from myocardial infarction or ischemic stroke [50]. However, even before thrombus formation, activated platelets can influence atherogenesis and atheroprogression, increasing the risk of future fatal thrombotic complications [51].

Ox-LDL-laden platelets not only induce endothelial inflammation, promoting vascular injury, but also inhibit the regeneration of the endothelium by reducing CD34^+^ progenitor cell differentiation into endothelial cells. Both of these processes promote atherogenesis initiation [52]. Platelets also influence atherosclerotic plaque development and destabilization by increasing lipid accumulation, monocyte migration, and foam cell formation (the key mediator of this process being the interaction between platelet P-selectin and P-selectin glycoprotein ligand-1 present on monocytes) [30]. Platelets also shape the immune response by releasing chemokines such as CXCL4, CCL5, and CXCL12 [53]. Moreover, ox-LDL-laden platelets can be phagocytosed by foam cells, directly increasing their lipid load [54].

Additionally, activated platelets can promote the activation of other platelets, increasing the atherogenic effect even more. Activated platelets can accelerate LDL-C oxidation through the generation of oxidative stress by platelet NADPH-oxidase 2, which further enhances platelet activation [55].

#### Platelet-Derived Extracellular Vesicles

Activated platelets release platelet-derived extracellular vesicles (PEVs) into the bloodstream, which can further increase platelet hyperreactivity [56,57]. EVs are membrane-enclosed mediators of cell–cell communication that are generated by various cells both in physiological and pathological states, and are heterogeneous both in terms of biogenesis and composition. PEVs constitute about 30% of all EVs detected in normal plasma [58].

PEVs may contain cyclooxygenase and thromboxane synthase, which can synthesize thromboxane, and thus promote platelet activation and aggregation. Moreover, PEVs can be used as substrates for the synthesis of arachidonic acid by phospholipase A2s, which in turn can be metabolized into thromboxane [59]. PEVs also contain various proteins characteristic of activated platelets and can therefore disseminate platelet activation [16]. It was shown that PEVs significantly increase fibrin deposition and platelet adhesion to the damaged vessel walls [60]. All these factors are responsible for the prothrombotic properties of PEVs.

PEVs can also promote inflammatory cytokine release and ox-LDL phagocytosis by macrophages, thereby accelerating foam cell formation and AS progression [61]. PEV interactions with the immune system and the resulting exacerbation of inflammation and oxidative stress can promote progression and destabilization of the atherosclerotic plaque and ox-LDL synthesis on various stages of atherosclerosis [16,62]. Additionally, PEVs may influence the adhesion of inflammatory cells and endothelial dysfunction, thus playing a part in the initial stages of atherosclerosis [16,63].

## 3. Markers of Platelet Activation in PH

Platelet activation can be detected through several markers, such as: (i) mean platelet volume, (ii) circulating PEV concentrations, (iii) platelet-derived inflammatory biomarkers, (iv) platelet-leukocyte aggregates, and (v) platelet-activating factor acetylhydrolase.

### 3.1. Mean Platelet Volume

One of the oldest, but still valuable, markers of platelet activation is mean platelet volume (MPV). Increased MPV is observed in various diseases and is associated with an increase in platelet activity and inflammation [64]. It has been shown that elevated MPV is associated with higher cardiovascular risk [62], however it should not be used as a standalone marker [64]. In PH, not only is MPV increased, but it is also independently associated with total cholesterol level [65].

### 3.2. Circulating PEV Levels

Liquid biopsy of circulating EVs may be a useful method for detecting atherosclerotic plaques and calcification in asymptomatic PH. In a study on eighty-two PH patients, the patients with atherosclerosis were characterized by higher levels of PEVs, regardless of lipid-lowering therapy. Combining PEV count with levels of other EVs resulted in 79.1% sensitivity and 45.8% specificity in detecting the presence of atherosclerotic plaques in PH patients [16]. Moreover, high levels of PEVs are present in young patients with high cardiovascular risk and are not completely normalized by lipid-lowering treatments [66]. Changes in the composition of EVs can also anticipate clinical events [67].

### 3.3. Platelet-Derived Inflammatory Biomarkers

Platelet factor 4 (PF4)/CXCL4, neutrophil activating peptide 2 (NAP2)/CXCL7, cluster of differentiation 40 ligand (CD40L), and regulated on activation normal T cell expressed and secreted (RANTES)/CCL5 were all found to be elevated in patients with FH, even in cases of intensive lipid-lowering treatment. This shows that despite platelet hyperactivation in FH being associated with elevated LDL-C levels, lowering blood LDL-C may not prevent all complications caused by platelet hyperreactivity [68].

### 3.4. Platelet–Leukocyte Aggregates

Platelets from PH patients express increased amounts of surface proteins such as P-selectin, resulting in a significantly higher tendency to create platelet–leukocyte aggregates. Not only is this a marker of the activation of platelets and leukocytes cells, but the presence of such aggregates also results in increased platelet and leukocyte adhesion to dysfunctional endothelium [69]. Platelet–leukocyte aggregate count is considered to be one of the most sensitive markers of platelet activation [70].

### 3.5. Platelet-Activating Factor Acetylhydrolase

Another platelet-associated marker that reflects the severity of hypercholesterolemia is platelet-activating factor acetylhydrolase (PAF-AH). It has been shown that the ratio of HDL-associated to LDL-associated PAF-AH decreases progressively from healthy to heterozygotic FH to homozygotic FH patients, and is proportional to the plasma LDL-C increase [71].

## 4. PCSK9 and PCSK9 Inhibitors

### 4.1. The Role of PCSK9

Expressed primarily in the liver, PCSK9 plays a key regulatory role in lipid metabolism [72,73]. As LDLR binds the LDL particle, the whole complex enters the endosomal pathway, eventually causing the degradation of LDL and releasing the LDLR back to the cell membrane [73]. PCSK9 binds to the LDLR on the cell surface, causing its internalization and lysosomal degradation [73]. This mechanism inhibits LDLR recycling, which normally allows one LDLR particle to process approximately 150 LDL particles [74,75].

Previous research reveals that PCSK9 overexpression is also regulated by non-genetic mechanisms [76]. Experimental data show that PCSK9 is induced by various inflammatory stimuli, such as lipopolysaccharides and zymosan, resulting in a significant increase in LDL-C levels [77]. Furthermore, ox-LDL also increases PCSK9 expression through the alteration of inflammatory cytokines such as interleukin (IL)-1α, IL-6, and tumor necrosis factor α (TNF-α) in macrophages [78]. This results in the progression of atherosclerosis, which involves platelets [17]. Interestingly, the siRNA-mediated knockdown of PCSK9 suppresses ox-LDL-induced proinflammatory chemokine synthesis [78].

### 4.2. PCKS9 and Platelets

There are many other receptors targeted by PCSK9 other than LDLR, such as CD36, low density lipoprotein receptor-related protein 1 (LRP-1), very low density lipoprotein receptor (VLDLR), and the apolipoprotein E receptor 2 (ApoER2) [79]. PCSK9 enhances platelet activation by binding to CD36, therefore contributing to atherosclerosis [79]. Besides lowering LDL-C level, PCSK9 inhibitor therapy showed a reduction in platelet reactivity and increased platelet sensitivity to the inhibitory effects of aspirin [24].

### 4.3. PCSK9 Inhibitors

Due to its function in lipid homeostasis, PCSK9 is a highly desirable target for therapeutic agents. Recently, a new class of drugs, PCSK9 inhibitors, has become available. The three members of this group available for patients in Europe are alirocumab, evolocumab and inclisiran.

Alirocumab and evolocumab are monoclonal antibodies (mAbs) that have been developed to bind PCSK9 and thus impair its function [80]. Clinical data show that the administration of PCSK9 mAbs is associated with an approximately 60% reduction in plasma LDL-C level in patients with both heterozygous FH and non-familial PH [81,82,83,84]. Anti-PCSK9 mAbs are injected subcutaneously. No major side effects have been described, yet there is the potential problem of autoantibodies [85]. Both alirocumab and evolocumab are fully human antibodies, and thus they are less likely to provoke such a reaction. However, few such incidents have been reported (without impairing the LDL-C lowering effect) [85].

Inclisiran is a relatively new drug that was authorized for use by the European Medicines Agency in December 2020. It is a silencing RNA (siRNA) particle targeting the hepatic production of PCSK9 [86]. Inclisiran selectively interferes with the expression of specific genes and catalytically silences the translation of the complementary target messenger RNA (mRNA), blocking the synthesis of PCSK9 [86]. Clinical trial data showed a 44% reduction in LDL-C level compared to placebo in heterozygous FH, and a 50% reduction in general hypercholesterolemic patients [87,88]. In contrary to mAbs, inclisiran needs to be administered twice a year, which is more convenient for patients than the twice-a-month injection of mAbs [86].

The indications for PCSK9-inhibitors include: (i) PH (heterozygous familial and non-familial) or mixed dyslipidaemia, (ii) homozygous FH, and (iii) established cardiovascular disease, in combination with diet and other lipid-lowering therapies [85]. Although PCSK9 inhibitors and inclisiran have great cholesterol-lowering potential in these patient populations, due to their novelty and high costs, they remain out of most patients’ reach [85].

## 5. Antiplatelet Effects of PCSK-9 Inhibitors

### 5.1. PCSK9 Inhibitors

The complex role of PCSK9 suggests that the impact of PCSK9 inhibition is not limited to the reduction of LDL-C, but that it also affects other aspects of PCSK9 activity, such as lipid metabolism and platelet function [89,90]. Moreover, as ox-LDL is a crucial factor for increasing platelet hyperreactivity, LDL-lowering treatment also affects platelets [91]. Until now, it has not been established whether PCSK9 inhibitors exert a direct inhibitory effect on platelet function, or whether this effect is secondary to the strong lipid-lowering potential of PCSK9 inhibitors [80,81,92].

In animal models, administration of the PCSK9-surpressing agent 10-dehydrogingerdione decreased both PSCK9 level and the concentration of platelet activation markers, such as soluble CD40 ligand and soluble P-selectin [93]. Concurrently, PCSK9 deficiency has been reported to attenuate thrombosis in mice [94]. Two studies conducted on small groups of patients receiving PCSK9 inhibitors in monotheraphy showed reduced platelet reactivity [24,95], further supporting the antiplatelet effects of PCSK9 inhibitors. No adverse effects on platelet counts were reported in patients receiving inclisiran [96]. Although the preliminary results are promising, there is still a lack of evidence-based data to draw firm conclusions regarding the mechanisms and magnitude of action of PCSK9 inhibitors, especially in monotherapy. For example, the effect of PCSK9 inhibitors on the concentrations of prostacyclin or thromboxane A_2_ and on platelet lifespan is still unknown, indicating that further research is needed to shed more light on this topic.

Besides the antiplatelet effects, higher PCSK9 levels were shown to accelerate the development of atherosclerotic plaques and increase the size of plaque necrotic cores, independent of lipid changes [97,98]. PCSK9 not only promotes ox-LDL internalization, both through interaction with LOX-1 and the increase in LDL level, but it also sensitizes cells to ox-LDL, which aggravates ongoing inflammatory processes. PCSK9 also stimulates dendritic cell maturation, which can in turn induce PCSK9, and T-cell proliferation [98]. Treatment with PCSK9 inhibitors was shown to: (i) decrease the formation of foam cells; (ii) inhibit the production of pro-inflammatory cytokines, including IL-1α, IL-6, and TNF-α, and the activation of proinflammatory pathways, such as the TLR4/NF-κB/COX-2 pathway; and (iii) suppress the migration and proliferation of smooth muscle cells [98,99,100]. PCSK9 inhibitors also decrease serum levels of cytokines associated with endothelial activation and monocyte/macrophage migration [99]. A human study showed that even a short-term therapy with PCSK9 inhibitors improves endothelial function, which is proportional to LDL reduction [101].

### 5.2. Statins and PCSK9 Inhibitors

Statin therapy is the first line of treatment for PH [85]. However, despite the high effectiveness of statins, there is considerable variability in the individual treatment response [102]. Even maximum doses may not achieve blood LDL-C targets, especially in patients with particularly high pre-treatment LDL-C levels. If the treatment goal is not achieved through therapy with statins and ezetimibe, treatment with a PCSK9 inhibitor is recommended [85].

It was found that PCSK9 polymorphisms can influence the therapeutic effect of statins [103]. It has also been suggested that slightly higher pre-treatment serum levels of PCSK9 may distinguish patients who do not respond to stain treatment [104]. Furthermore, it was found that statin therapy causes a significant increase in plasma PCSK9 concentration [105]. It has been suggested that this might be the reason for the nonlinear relationship between statin dose and LDL-C reduction, where at some point, increasing the statin dose does not exert additional effects on LDL-C levels [106].

Monotherapy with statins reduces platelet activation and inflammation [107]; however, this effect is significantly correlated with LDL-C reduction [91]. In contrast, PCSK9 inhibitors were found to reduce platelet reactivity, both directly and through their LDL-lowering effect [24]. They also promote atherosclerotic plaque stabilization [108]. As a result, adding PCSK9 inhibitors to statin therapy significantly improves cardiovascular outcomes, both due to the lipid-lowering and antiplatelet effects [109,110,111].

### 5.3. Ezetimibe and PCSK9 Inhibitors

Adding ezetimibe to statin therapy can further reduce LDL-C levels and improve endothelial function [112], but the LDL-lowering effect of ezetimibe is lower than that of PCSK9 inhibitors [113]. Ezetimibe also significantly reduces the expression of P-selectin and CD40L on the surface of free platelets; however, this effect was not observed in platelets in direct contact with endothelial cells [114].

Ezetimibe targets microsomal triglyceride transfer protein (MTP) and NPC1L1, which are upregulated as a result of PCSK9 increase [115], which occurs during both ezetimibe and statin therapy, and this effect is increased even more when both drugs are used simultaneously [116]. Therefore, ezetimibe and PCSK9 inhibitors work synergistically and both can be seen as a complement of statin therapy [85,115].

### 5.4. Antithrombotic Therapy and PCSK9 Inhibitors

Antithrombotic therapy is a crucial part of atherosclerosis treatment [117], and pre-clinical trials show that it is also useful before the late stage of hypercholesterolemia, as a method of inhibiting the hypercholesterolemia–inflammation loop to stop the initiation and progression of atherosclerosis [118]. This effect is better documented in the case of P2Y12 inhibitors, clopidogrel and ticagrelor, than acetylsalicylic acid (ASA) [117]. A study on ticagrelor showed that its atherosclerosis-alleviating effect may be linked to PSCK9 downregulation [119].

However, in up to 30% of patients, platelets show decreased sensitivity to antithrombotic therapy. It has been shown that the suboptimal response to ASA can be linked to hypercholesterolemic states, and that lipid lowering therapy may improve the response [120]. A similar correlation has been found between clopidogrel responsiveness and LDL-C level [121]. In the case of prasugrel and ticagrelor, PCSK9 levels have been shown to correlate with platelet activity during treatment [122]. Therefore, combining antithrombotic therapy with PCSK9 inhibitors can have beneficial effects both on platelet activity and atherosclerosis progression.

A summary of the studies investigating the effects of PCSK9 inhibitors or PCSK9 levels on platelet function parameters is shown in Table 1.

## 6. Conclusions

Lifelong exposure to elevated LDL-C levels in PH result in the early onset of atherosclerosis, the development of which is significantly influenced by platelet hyperactivation. Platelets are therefore attractive therapy targets and subduing their activation can be achieved by simply lowering blood LDL-C levels. However, many PH patients fail to achieve cholesterol level goals using current widespread therapies. For such patients, PCSK9 inhibitors are a promising way of normalizing LDL-C levels and decreasing the platelet-derived oxidative and inflammatory burden that has built up over the years. They can also be a valuable addition to traditional lipid-lowering and antithrombotic therapies, increasing their effectiveness in many ways. It must, however, be noted that PCSK9 inhibitors are relatively new drugs, and therefore further research is needed to assess their long-term effects on atherosclerosis development and to find the most effective treatment strategies.

## Figures and Tables

**Figure 1 life-11-00466-f001:**
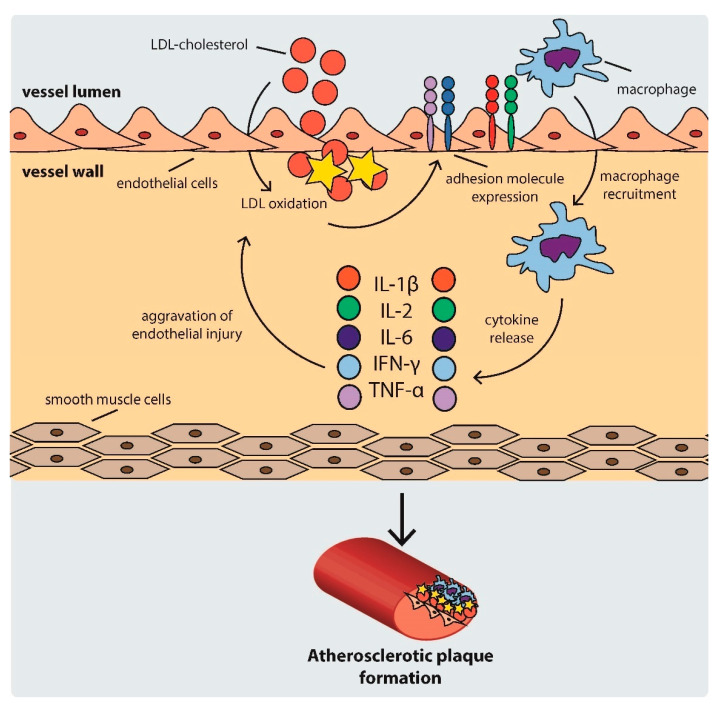
Scheme of atherosclerotic plaque development. IFN: interferon, IL: interleukin, LDL: low density lipoprotein, TNF-α: tumor necrosis factor α.

**Figure 2 life-11-00466-f002:**
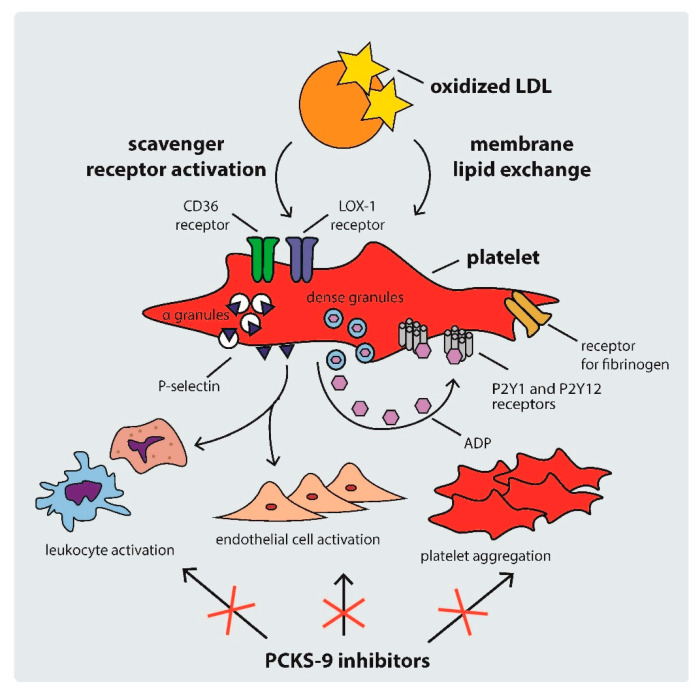
Mechanisms of platelet activation by LDL. ADP: adenosine diphosphate, CD: cluster of differentiation, LDL: low density lipoprotein, PCSK-9: proprotein convertase subtilisin-kexin type 9.

**Table 1 life-11-00466-t001:** Summary of studies investigating the effects of PCSK9 inhibitors or PCSK9 levels on platelet function parameters. PH: primary hypercholesterolemia, FH: familial hypercholesterolemia, LDL-C: low-density lipoprotein cholesterol, sP-selectin: soluble P-selectin, sCD40L: soluble CD40 ligand.

PCSK-9 Inhibitor	Population	Effect	Ref.
Monotherapy			
Alirocumab/evolocumab	Patients with PH (n = 7)	Decrease in P-selectin exposure, with and without agonists	[24]
Alirocumab/evolocumab	Patients with hypercholesterolemia (n = 21)	Reduced platelet reactivity to agonists	[95]
Alirocumab	Patients with FH (n = 736)	LDL-C lowering	[81]
Evolocumab	Patients with FH (n = 331)	LDL-C lowering	[92]
10-Dehydrogingerdione	Rabbits (n = 30)	Decrease in sP-selectin and sCD40L	[93]
PCSK9 deficiency	Mice (n = 20)	Lowered risk of venous thrombosis	[94]
Polytherapy			
Alirocumab + statin (unspecified)	Patients with hypercholesterolemia (n = 18,924)	Decreased risk of thrombotic events	[110]
Evolocumab + statin (unspecified)	Patients after acute coronary syndrome (n = 18,924)	Decreased risk of venous thromboembolism	[111]
Evolocumab + rosuvastatin	Patients with de novo acute coronary artery disease (n = 64)	Stabilization of atherosclerotic plaque	[108]
Loss-of-funcion mutation in PCSK9 gene + statin (unspecified)	Patients with hypercholesterolemia (n = 2388)	Improved response to statin therapy	[103]
Alirocumab/evolocumab + aspirin	Patients with PH (n = 14)	Decrease in P-selectin exposure, with and without stimuli	[24]
Alirocumab + aspirin	In vitro study (n = 10)	Decrease in platelet aggregation	[79]
Lower levels of PCSK9 + ticagrelor	Patients with acute coronary syndrome (n = 333)	Decrease in platelet aggregation	[122]

## Data Availability

Not applicable.

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
