# Peer review of "Antiplatelet Effects of PCSK9 Inhibitors in Primary Hypercholesterolemia"

_life, 2021, doi:10.3390/life11060466_

Round 1

Reviewer 1 Report

The authors have investigated and summarized the platelet hyperreactivity in PH and the effects of PCSK9 inhibitors on platelets. There are some points to note. They described that lipid disorder involves both autosomal-dominant FH and the more frequent polygenic non-FH in line 35-38. Recently, clinical-FH includes monogenic and polygenic causes. What kind of patients is “more frequent polygenic non-FH” intended for? In Figure 1, the reviewer suggests that it is easy to understand that the relationship between macrophages and vascular endothelial cells and the relationship of oxLDL with each cell are illustrated as events that occur in blood vessels. In addition, the behavior of LDL in cells should be described as LDL instead of LDL-C in line 77, 83 etc. In line 354-355, target of ezetimibe is generally NPCL1. If the expression of MTP and NPCL1 is up-regulated by increasing PCSK9, reference should be shown. In Figure 2, is there any evidence that PCSK9 inhibitor inhibits leukocyte activation and endothelial cell activation? If the authors have proof of that, they should mention it in “Antiplatelet effects of PCSK9-inhibitors” section.

Author Response

Dear Reviewer,

we are thankful for the time and effort that you spent to provide in-depth review of our review paper. We corrected our manuscript according to your suggestions. Our response and corrections are listed below.

The authors have investigated and summarized the platelet hyperreactivity in PH and the effects of PCSK9 inhibitors on platelets. They described that lipid disorder involves both autosomal-dominant FH and the more frequent polygenic non-FH in line 35-38. Recently, clinical-FH includes monogenic and polygenic causes. What kind of patients is “more frequent polygenic non-FH” intended for?

We thank the Reviewer for this question. We clarified the distinction between the monogenic and polygenic hypercholesterolemia in the text. Specification of each variant of hypercholesterolemia had been described in line 42-51 of the original manuscript, as follows: Three main genes have been identified as causative factors of FH in an autosomal dominant manner: the LDL receptor (LDLR), apolipoprotein B (ApoB) and the pro-protein convertase subtilisin-kexin type 9 (PCSK9) [10]. There are other rare forms of FH as well. These include the mutations in the apolipoprotein E (ApoE) gene and LDL-C adaptor protein 1 (LDLRAP1) [11, 12]. The polygenic non-familial hypercholesterolemia variant includes single nucleotide polymorphisms (SNPs) in several genes, involving the common ones (LDLR, PCSK9) as well as less frequent genes, such as ATP-binding cassette sub-family G member 8 gene (ABCG8) or Cadherin EGF LAG seven-pass G-type receptor 2 gene (CELSR2) [13]. Both ABCG8 and CELSR2 code proteins associated with transmembrane transport and receptor-ligand cellular interactions [14, 15].  

In Figure 1, the reviewer suggests that it is easy to understand that the relationship between macrophages and vascular endothelial cells and the relationship of oxLDL with each cell are illustrated as events that occur in blood vessels.

We thank the Reviewer for raising this point. We modified the Figure showing that these interactions occur in the vessel wall.

In addition, the behavior of LDL in cells should be described as LDL instead of LDL-C in line 77, 83 etc.

We corrected our manuscript according to your suggestions.

In line 354-355, target of ezetimibe is generally NPCL1. If the expression of MTP and NPCL1 is up-regulated by increasing PCSK9, reference should be shown.

We thank the Reviewer for pointing this out. The citation for this claim was provided at the end of the next sentence (number 109). In the revised version we provided the citation also in the sentence about MTP and NPCL1, and additionally slightly modified the part of the sentence about the increase of PCSK9 during therapy, as follows “(…) [PCSK9 increase] occurs during both ezetimibe and statin therapy, and this effect is increased even more when those drugs are used simultaneously” and added another citation.

In Figure 2, is there any evidence that PCSK9 inhibitor inhibits leukocyte activation and endothelial cell activation? If the authors have proof of that, they should mention it in “Antiplatelet effects of PCSK9-inhibitors” section.

We thank the Reviewer for noticing this. We added an explanation of the role of PCSK9 and its inhibitors in these processes to the text, as follows “Additionally, it has been shown that higher PCSK9 levels can accelerate the development of atherosclerotic plaque and increase the size of plaque necrotic core, independent of lipid changes [97, 98]. PCSK9 not only promotes oxLDL internalization, both through interaction with LOX-1 and the increase in LDL level, but it also sensitizes the cells to oxLDL, which aggravates the ongoing inflammatory processes. PCSK9 also stimulates dendritic cell maturation, which can in turn induce PCSK9, and T-cell proliferation [98]. Treatment with PCSK9 inhibitors was showed to (i) decrease the formation of foam-cells, (ii) inhibit the production of pro-inflammatory cytokines, including IL-1α, IL-6, and TNF-α, and activation of proinflammatory pathways, such as TLR4/NF-κB/COX-2 pathway, and (iii) suppresses migration and proliferation of smoth muscle cells [98-100]. PCSK9 inhibitors also decrease serum levels of cytokines associated with endothelial activation and monocyte/macrophage migration [99]. A human study showed that even a short-term therapy with PCSK9 inhibitors improves endothelial function, which is proportional to LDL reduction [101].

Altogether, we are grateful for the in-depth revision of our manuscript and we hope that it will be considered for publication.

On behalf of all Authors,

Sincerely,

Mateusz Leśniewski

Piotr Pęczek

Aleksandra GÄ…secka

Reviewer 2 Report

The topic of this paper is novel, dealing with novel iPCSK9 drugs and their pleiotropic properties. This is a review paper, very well written, well selected literature. No editorial remarks. In my opinion, the paper should be published as soon as possible, because it indicates important additional antiplatelet properties of these drugs, extremely important in patients after acute myocardial infarction.

Author Response

The topic of this paper is novel, dealing with novel iPCSK9 drugs and their pleiotropic properties. This is a review paper, very well written, well selected literature. No editorial remarks. In my opinion, the paper should be published as soon as possible, because it indicates important additional antiplatelet properties of these drugs, extremely important in patients after acute myocardial infarction.

Dear Reviewer,

we are thankful for the time and effort that you spent to provide in-depth review of our review paper. We are grateful for appreciating our work.

On behalf of all Authors,

Sincerely,

Mateusz Leśniewski and Piotr Pęczek

Reviewer 3 Report

There have been many reports concerning PCSK2 inhibitors can reduce blood LDL-C. However, not many studies on the inhibition of platelet aggregation by PCSK9 inhibitors were reported. There is still no evidence to show whether it is a direct or indirect effect. Therefore, the mechanism is unclear. This is an interesting study. The manuscript may provide some possible mechanisms and research directions, and it is somehow worthy of reference. But it is still lacking of strong evidence base, including the changes of platelet number, platelet lifespan, PGI2, TXA2 and vascular endothelial cells after PCSK2 inhibitors treatment.

Author Response

Dear Reviewer,

we are thankful for the time and effort that you spent to provide in-depth review of our review paper. We corrected our manuscript according to your suggestions. Our response and corrections are listed below.

There have been many reports concerning PCSK2 inhibitors can reduce blood LDL-C. However, not many studies on the inhibition of platelet aggregation by PCSK9 inhibitors were reported. There is still no evidence to show whether it is a direct or indirect effect. Therefore, the mechanism is unclear. This is an interesting study. The manuscript may provide some possible mechanisms and research directions, and it is somehow worthy of reference. But it is still lacking of strong evidence base, including the changes of platelet number, platelet lifespan, PGI2, TXA2 and vascular endothelial cells after PCSK2 inhibitors treatment.

We thank the Reviewer for appreciating our paper. We are aware PCSK9 inhibitors are relatively new pharmacological agents and thus their pleiotropic effects have not yet been fully researched. This applies also to their effects on the platelets. However, we believe that that our paper will help draw attention to the association between platelet function and PCSK9 inhibition, that we think is an important aspect of lipid-lowering therapy.

We added several sentences about the role of PCSK and its inhibitors on inflammation and endothelial cells, as follows: Additionally, it has been shown that higher PCSK9 levels can accelerate the development of atherosclerotic plaque and increase the size of plaque necrotic core, independent of lipid changes [97, 98]. PCSK9 not only promotes oxLDL internalization, both through interaction with LOX-1 and the increase in LDL level, but it also sensitizes the cells to oxLDL, which aggravates the ongoing inflammatory processes. PCSK9 also stimulates dendritic cell maturation, which can in turn induce PCSK9, and T-cell proliferation [98]. Treatment with PCSK9 inhibitors was showed to (i) decrease the formation of foam-cells, (ii) inhibit the production of pro-inflammatory cytokines, including IL-1α, IL-6, and TNF-α, and activation of proinflammatory pathways, such as TLR4/NF-κB/COX-2 pathway, and (iii) suppresses migration and proliferation of smoth muscle cells [98-100]. PCSK9 inhibitors also decrease serum levels of cytokines associated with endothelial activation and monocyte/macrophage migration [99]. A human study showed that even a short-term therapy with PCSK9 inhibitors improves endothelial function, which is proportional to LDL reduction [101]. All of those claims are supported with suitable citations.

We also added the information about changes in platelet number as follows: “No adverse effects on platelet counts were reported in patients receiving inclisiran”. However, we could not find any research investigating the effects of PCSK9 on PGI2, TXA2 concentrations and  platelet lifespan, which we added to our manuscript as follows: “Although the preliminary results are promising, there is still a lack of evidence-based data to draw firm conclusions regarding the mechanisms and magnitude of PCSK9 inhibitors, especially in monotherapy. For example, the effect of PCSK9 inhibitors on the concentrations of prostacyclin or thromboxane A2 and on platelet lifespan is still missing, indicating that further research is needed to shed more light on this topic.”

Altogether, we are grateful for the in-depth revision of our manuscript and we hope that it will be considered for publication.

On behalf of all Authors,

Sincerely,

Mateusz Leśniewski,

Piotr Pęczek,

Aleksandra GÄ…secka

Round 2

Reviewer 1 Report

The authors responded approximately to the reviewer's comments. The manuscript was improved by revision.

Reviewer 3 Report

This manuscript is now suitable for publication in this journal.